# Development of SSR markers related to salinity resistance based on transcriptomic sequences in *Medicago sativa*

Rugang Yu[1,2], Xin Chen[1], Hui Zhang[1], Qiting Zhang[1], Xinyi Chen[1], Yanqiu Dong[1], Liwei Chen[1], Daniel Basigalup[3], Guoliang Wang[3*], Xueling Du[1,2*]

**1** College of Life Sciences, Huaibei Normal University, Huaibei, Anhui, China, **2** Anhuii Province Key Laboratory of Pollutant Sensitive Materials and Environmental Remediation, Huaibei, Anhui, China, **3** Institute of Leisure Agriculture, Shandong Academy of Agricultural Sciences, Jinan, Shandong, China

* wangguoliang@126.com (GW); duxueling1208@126.com (XD)

## Abstract

Alfalfa (*Medicago sativa*) is an important perennial forage crop that exhibits wide cultivar variations in salinity tolerance. Simple sequence repeats (SSRs) in a transcriptome can realize targeted markers that are directly related to target traits. However, SSR markers related to specific traits, especially salinity tolerance traits in alfalfa, are rarely reported worldwide. This study aimed to investigate the distribution characteristics of SSR loci and explore the key SSR loci related to salinity-tolerant genes in alfalfa. For this purpose, we conducted transcriptomic analysis of roots and leaves from GIB (G, high salinity-tolerant) and LS (L, high salinity-sensitive) plants under 0 and 200 mM NaCl treatments, which yielded 129,563 unigenes. A total of 38,370 SSR loci were identified and distributed in 28,039 unigenes, and the frequency of SSR occurrence in each locus was 4.43 kb. Among all the SSR motifs, mononucleotide (67.32%), trinucleotide (15.61%), and dinucleotide (14.53%) were the major repeated types, and the forms of A/T, AG/CT, AAG/CTT, AC/GT, AT/AT and AAC/GTT were the most frequent motifs. Meanwhile, 23,159 primer pairs of SSRs were designed for marker development in alfalfa. Among the 28,039 SSR-containing unigenes, 1,947 unigenes were found to be salinity-responsive differentially expressed unigenes (DEUs) and/or DEUs between the two cultivars. Interestingly, 188 DEUs were identified and found to be involved in ion transport, metabolite biosynthesis, ROS regulation, signaling pathway, and transcription regulation, which were all related to salinity tolerance. Notably, six out of 211 SSR loci identified based on 188 SSR-containing DEUs were validated as polymorphic SSR markers with clear amplified bands, which they exhibited high polymorphism (polymorphism information content: 0.640–0.807). Therefore, these SSR markers could be further used for authenticity identification and genetic analysis. The six SSRs were used to classify four alfalfa varieties with different salinity tolerance into three groups. The high salinity-sensitive variety LS was placed in group I, the high tolerant varieties GIB and GN5 formed group II, and

**Data availability statement:** The raw transcriptomics data generated in this study are openly available in GSA in the CNCB with accession number PRJCA019338 (https://ngdc.cncb.ac.cn/gsa). All data underlying this article are included in the article and its online supplementary files.

**Funding:** This study was financially supported by the Earmarked Fund for China Agriculture Research System in the form of a grant (CARS-34) received by GW. This study was also financially supported by the Key Research and Development Program of Shandong Province in the form of a grant (WSR2023050) received by GW. This study was also financially supported by the National College Student Innovation and Entrepreneurship Training Program in the form of a grant (202410373017) received by XD. This study was also financially supported by the Huaibei Normal University Re-established Project in the form of a grant (2024ZK38) received by RY.

**Competing interests:** The authors have declared that no competing interests exist.

the sensitive variety GN3 was included in group III. This grouping was consistent with prior evaluations of salinity tolerance. Therefore, the six SSRs may be associated with salinity tolerance in alfalfa. These findings not only provide an efficient tool for the large-scale development of markers related to specific traits but also lay a foundation for genetic analysis in alfalfa.

## Introduction

Alfalfa (*Medicago sativa* L.), which is a member of the Fabaceae family, is a forage crop cultivated worldwide; it is renowned as the "Queen of forages" because of its superior nutritive value and functional properties [1,2]. Soil salinity is among the critical environmental problems influencing alfalfa yield [3]. The screening and breeding of alfalfa varieties with salinity tolerance is critical for improving and utilizing salinized soil. Salinity tolerance in alfalfa has been known as a complex quantitative trait that is regulated by numerous genes. Notably, a significant variability in salinity tolerance exists among alfalfa varieties [3–6]. Therefore, the salinity tolerance among alfalfa varieties needs to be explored. However, the intricate salinity tolerance mechanism of alfalfa and its allogamous autotetraploid [7] cause difficulty in distinguishing the salinity tolerance among varieties.

Molecular markers are derived from defined sites within genes or regulatory sequences of plant genomes, which can be used to discriminate between and within species at the DNA level [8] Among various marker types, simple sequence repeats (SSRs) have become a crucial tool in genetic studies because of their high reproducibility, high polymorphism, codominance, and easy amplification through polymerase chain reaction (PCR) [9,10]. SSRs are short tandemly repeated sequences with motifs ranging from 1 to 6 base pairs (bp), which are distributed in coding and non-coding regions throughout the plant genomes [11]. In plants, SSR markers have been extensively employed in genetic diversity assessment [12,13], germplasm resource identification and classification [14], and trait association analysis [10,15]. For example, Li et al. utilized 15 SSR markers to identify the genetic diversity of 25 *Ilex asprella* resources, which could be divided into three populations [12]. Nie et al. used 30 randomly selected SSRs to perform marker–trait association analysis and identified 7 significant associations with drought tolerance-related traits in *Miscanthus* [15]. In alfalfa, the application of SSR markers focuses primarily on the analysis of genetic diversity [16] and identification of germplasm resources [17]. However, SSR markers related to specific traits, especially salinity tolerance trait in alfalfa, are rarely reported worldwide.

SSRs can be divided into two categories, namely, genomic SSRs and expressed sequence tag SSRs, which originate from genomic and transcriptomic sequences, respectively [10,18]. Compared with the traditional mining SSRs from genomic sequences, the developed SSR markers from transcriptomic sequences were considered a simple and effective method with low cost and less labor [14,19]. Currently, the discovery and mining of SSR loci through transcriptomic sequences have been successfully applied to many plant species [20,21]. For instance, in *Auricularia heimuer*, 53

polymorphic EST-SSRs were identified from transcriptome sequencing, of which 13 SSR markers were employed to analyze the genetic relationships of 52 *A. heimuer* germplasms [20]. In *Fagopyrum esculentum*, 20,756 EST-SSR loci within the transcriptomic sequences were detected, and 224 SSR primer pairs employed them to identify the genetic relationships of 48 *F. esculentum* varieties [21]. However, mining of SSR markers through transcriptomic sequences is rarely reported in alfalfa.

In this study, we developed SSR markers related to salinity tolerance by transcriptome sequencing. The primary aims of this study are to (i) explore the gene expression changes induced by NaCl in two alfalfa varieties with different salinity tolerance; (ii) identify the SSR-containing differentially expressed unigenes (DEUs) related to the salinity stress tolerance of alfalfa; (iii) develop SSR markers associated with salinity tolerance; and (iv) validate whether the developed SSR markers can be applied to identify salt tolerance traits in alfalfa. These findings will facilitate the development of SSR markers, genetic analysis, and molecular breeding studies in alfalfa.

## Materials and methods

### Plant materials and stress treatments

Our previous studies showed that Gibraltar (GIB) and Gannong No.5 (GN5) were high salinity-tolerant varieties, Gannong No.3 (GN3) was a salinity-sensitivity variety, and LS1405 (LS) was a variety of high salinity sensitivity [5,22]. Therefore, two varieties with significant differences in salt tolerance, namely, GIB and LS, were used for transcriptome sequencing to identify key genes related to salinity stress tolerance and develop SSR markers linked to this trait. In addition, two salinity-tolerant varieties (GIB, GN5) and two salinity-sensitive varieties (GN3, LS) were selected to validate the newly developed SSR markers. Seeds of GN5, GN3, LS, and GIB were procured commercially from the Institute of Leisure Agriculture (Jinan, China), Purple Posture Co., Ltd. (Bengbu, China), Beijing S&G Eco-Tech Co., Ltd. (Beijing, China), and Beijing Best Grass Industry Co., Ltd. (Beijing, China), respectively. In our experiment, pot trials were performed in a greenhouse at Huaibei Normal University (Huaibei, Anhui, China) under controlled conditions ($25 \pm 2$°C with a 16/8 h light/dark cycle). Seeds of the varieties were surface sterilized in a 0.1% $HgCl_2$ solution and then directly planted in sand-filled pots. After one week sowing, seedlings were thinned to eight uniform plants/pot irrigated with a 100 mL Hoagland nutrient solution (pH 5.8). Two weeks after sowing, the seedlings of GN5 and GN3 continued to be cultured with a nutrient solution, while the seedlings of GIB and LS were treated with a 100 mL nutrient solution supplemented with 0 and 200 mM NaCl, respectively. The nutrient solutions were replaced every two days. The NaCl concentrations were set according to the results of a preliminary experiment, which showed a significant difference in growth between GIB and LS under 200 mM NaCl treatment. Four weeks after sowing, root and leaf samples of GIB and LS were selected for transcriptome sequencing. Moreover, the leaf samples of GIB, GN5, GN3 and LS under NaCl-free treatment were selected to validate the newly developed SSR markers. All collected samples were flash-frozen in liquid nitrogen and stored at −80°C. Equal amount of root/leaf samples from four randomly selected individual plants (per pot) from each treatment were pooled and collected as a replicate. Two biological replicates were analyzed for transcriptome sequencing.

### Transcriptome sequencing and SSR loci mining

Total RNA was isolated from root and leaf tissues of GIB (G) and LS (L) alfalfa plants treated with 0 and 200 mM NaCl by using Trizol® Reagent (Invitrogen, Carlsbad, CA, USA) according to the instructions of the manufacturer. The total amounts and integrity of RNA were assessed using the Bioanalyzer 2100 system (Agilent, CA, USA). A total of 16 cDNA libraries of roots (i.e., $G_{0\_R1}$, $G_{0\_R2}$, $G_{200\_R1}$, $G_{200\_R2}$, $L_{0\_R1}$, $L_{0\_R2}$, $L_{200\_R1}$, and $L_{200\_R2}$) and leaves (i.e., $G_{0\_L1}$, $G_{0\_L2}$, $G_{200\_L1}$, $G_{200\_L2}$, $L_{0\_L1}$, $L_{0\_L2}$, $L_{200\_L1}$, and $L_{200\_L2}$) were constructed and sequenced on the Illumina NovaSeq 6000 platform (Novogene, Beijing, China). After sequencing, clean reads were procured from the raw reads by removing the adapter sequences, N bases, and low-quality reads. Then, using Trinity (v2.6.6) [23] software, the clean reads were assembled to the reference unigene sequences for further analysis. Finally, unigenes were functionally annotated by BLAST searches against NR, Nt, Pfam, SwissProt, GO, COG/KOG, and KEGG databases.

Potential SSR loci were mined using MISA 1.0 software with the following parameters: motif lengths ranging from 1 to 6 bp, and minimum repeat counts of 10 (mono-nucleotide repeat motifs), 6 (dinucleotide repeat motifs), and 5 (all other motif types).

### Identification and enrichment analysis of DEUs containing SSR loci

Read counts of each unigene were quantified by RSEM [24]. Then, read counts of SSR-containing unigenes based on a negative binomial distribution were used for differential expression analysis among root groups ($G_{0\_R}$, $G_{200\_R}$, $L_{0\_R}$, and $L_{200\_R}$) and leaf groups ($G_{0\_L}$, $G_{200\_L}$, $L_{0\_L}$, and $L_{200\_L}$) using the DESeq2 R package (1.20.0) [25]. DEUs were defined as those with $|\log_2 (\text{fold-change})| > 1$ and $P_{adj} < 0.05$. Functional enrichment of SSR-containing DEUs was analyzed using GOseq (v1.10.0) and KOBAS (v2.0.12) for GO and KEGG pathways ($P_{adj} < 0.05$).

### SSR primer design and PCR amplification

SSR primers were designed using Primer 3 (v2.5.5) software with the following criteria: a primer length of 18–27 bp, GC content of 40%–55%, annealing temperature (TM) ranging from 57°C to 62°C, a ΔTm (difference in annealing temperature between primers) of ≤ 5°C, and a PCR product size of 100–280 bp.

The SSR markers were validated as follows: (1) DNA was extracted from four alfalfa varieties with different salinity tolerance under NaCl-free treatment using M5 Plant Genomic DNA Kit (Mei5bio, Beijing, China), and an ultra-micro spectrophotometer (Aurora-900) was utilized for the DNA quantification; (2) 11 randomly selected SSR primer pairs (S1 Table) were used to verify the validity of the SSR primers by PCR with GIB material at DNA level; (3) SSR-containing DEUs related to salinity tolerance regulation were selected to develop markers related to salinity tolerance in alfalfa, and the developed SSR markers were validated by PCR with GIB, GN5, GN3, and LS materials at DNA level; (4) PCR amplification was conducted using 2X M5 HiPer plus Taq HiFi PCR mix (Mei5bio, Beijing, China) in a 10 μL reaction mixture, which consisted of 5 μL PCR Mix, 1 μL DNA (20 ng/μL), 0.5 μL forward primer (10 mM), 0.5 μL reverse primer (10 mM), and 3 μL ddH₂O; (5) PCR products were separated by 8% polyacrylamide gel electrophoresis and visualized by silver staining.

### Genomic localization of SSR-containing DEUs related to salinity tolerance

The cultivated alfalfa genomic sequence was downloaded from figshare (https://figshare.com/articles/dataset/genome_fasta_sequence_and_annotation_files/12327602) [26]. First, unigenes related to salinity tolerance were aligned to the alfalfa genome using local BLASTN implemented in BioEdit. Genes were retained if the alignment showed greater than 80% and an $e$-value $< 1e^{-50}$. The chromosomal localizations of the SSR-containing genes related to salinity tolerance were extracted from the GTF database and visualized using TBtools (v2.305) software.

### SSR data analysis

A 1–0 matrix dataset representing the presence (1) or absence (0) in the banding profile of SSR markers in four alfalfa varieties was constructed. Parameters related to genetic diversity (Na, Ne, H, and I) were computed using PopGene 32. The polymorphism information content (PIC) of each SSR marker was determined by PIC_CALC v.0.6. UPMGA cluster analysis was performed using NTSYS v.2.10e.

## Results

### RNA sequencing and assembly

The cDNA libraries constructed from the roots and leaves of two alfalfa varieties (NaCl/control RNA samples) were sequenced. Deep sequencing yielded between 21.27–22.84 million raw reads per library, of which 20.69–22.19 million were clean reads. Then, these clean reads were assembled into 129,563 unigenes with an average length of 959 bp and

an N50 length of 1,303 bp. Among them, 117,159 unigenes were annotated according to the public databases. The raw transcriptome data generated in this study are openly available in Genome Sequence Archive (GSA) in the China National Center for Bioinformation (https://ngdc.cncb.ac.cn/gsa, No.: PRJCA019338).

## Comprehensive SSR identification

A total of 28,039 unigene sequences containing 38,370 potential SSRs with a distribution frequency of 21.64% were identified from 129,563 unigene sequences (Table 1). Among them, 7,288 unigene sequences contained more than one SSR, and 2,962 SSRs were present in compound formation (Table 1). In addition, six types of SSRs, including mono-, di-, tri-, tetra-, penta-, and hexanucleotide repeat motifs, were identified from the transcriptome data. The frequency of SSRs based on the number of repeat motif differed (Table 2). Among them, mononucleotide (67.32%) repeats were the most abundant, followed by trinucleotide (15.61%), dinucleotide (14.53%), tetranucleotide (1.45%), hexanucleotide (0.55%), and pentanucleotide (0.54%) repeat units (Table 2).

A total of 196 different SSR motif types were identified from alfalfa transcriptome. Among them, A/T was the most abundant found motif, which accounted for 67.07%; AG/CT (7.28%), AAG/CTT (4.11%), AC/GT (3.71%), AT/AT (3.47%), and AAC/GTT (3.34%) were also abundant. The largest numbers of 28 SSR motif types of SSR in alfalfa transcriptome are shown in Fig 1.

## Design and validation of SSR primers

A total of 22,445 unigenes were successfully screened from 28,039 unigenes with SSR loci, and 23,159 primer pairs of SSRs were designed (S2 Table). Among them, 11 primer pairs (S2 Table) were randomly screened for PCR amplification. Among them, 9 SSR primer pairs successfully amplified clear bands in GIB variety, and the primer conversion rate was 81.82% (Fig 2). This result suggests that the SSR primers could be used for further analysis.

**Table 1. Results of SSR search in alfalfa transcriptome.**

| Iterm | Numbers |
|---|---|
| Total size of examined sequences/bp | 124,239,797 |
| Total number of identified SSRs | 38,370 |
| Frequency of occurrence/% | 21.64 |
| Number of SSR containing sequences | 28,039 |
| Number of sequences containing more than 1 SSR | 7288 |
| Number of SSRs present in compound formation | 2962 |

**Table 2. Frequency of SSR types in the alfalfa transcriptome.**

| Repeat motif types | Repeat numbers | | | | | | | Total | % |
|---|---|---|---|---|---|---|---|---|---|
| | 5 | 6 | 7 | 8 | 9 | 10 | >10 | | |
| Mo | 0 | 0 | 0 | 0 | 0 | 9335 | 16,495 | 25,830 | 67.32 |
| Di | 0 | 1578 | 861 | 623 | 439 | 321 | 1752 | 5574 | 14.53 |
| Tri | 3148 | 1329 | 610 | 327 | 243 | 132 | 202 | 5991 | 15.61 |
| Tetra | 327 | 145 | 52 | 8 | 7 | 3 | 14 | 556 | 1.45 |
| Penta | 153 | 38 | 3 | 6 | 2 | 2 | 5 | 209 | 0.54 |
| Hexa | 156 | 16 | 16 | 10 | 5 | 4 | 3 | 210 | 0.55 |
| Total | 3784 | 3106 | 1542 | 974 | 696 | 9797 | 18,471 | 38,370 | |
| % | 9.86 | 8.09 | 4.02 | 2.54 | 1.81 | 25.53 | 48.14 | | |

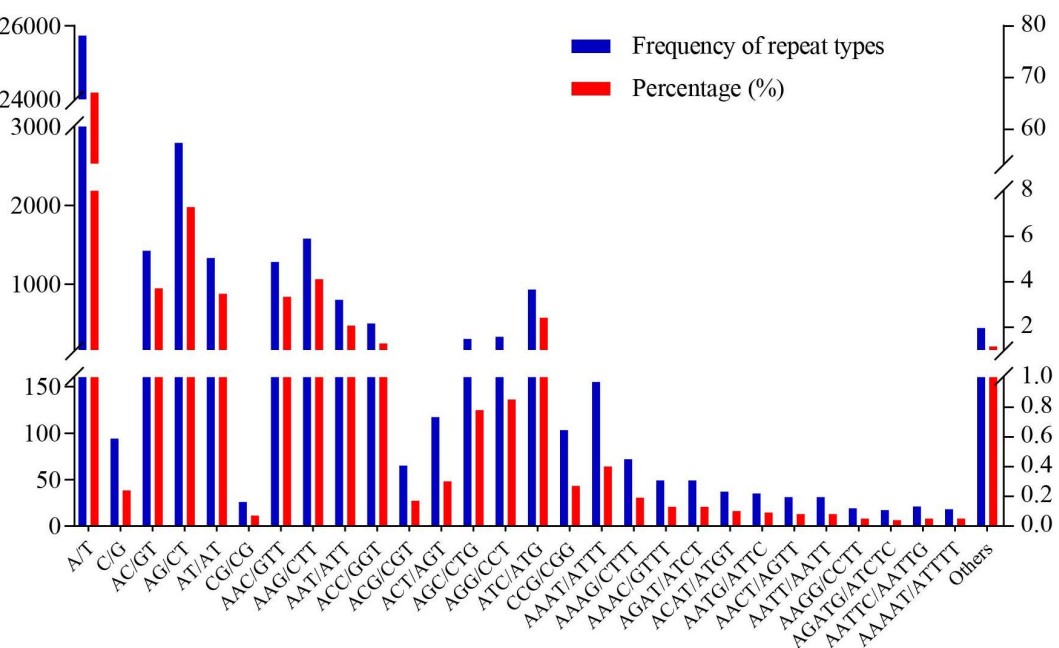

**Fig 1. Frequency distribution of SSRs based on motif types.**

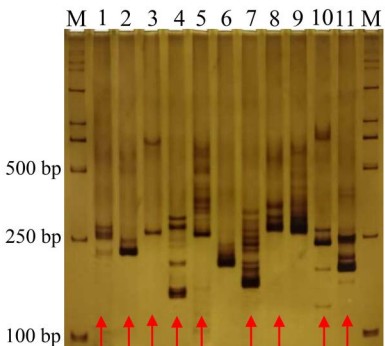

**Fig 2. Amplification results of 11 different primers in Gibraltar.** M: DL 2000 marker; the number 1–11 represent random primer 1–11; the red arrow indicates the target strip.

## Identification of SSR loci-containing DEUs

Pairwise comparison analysis for each unigene was conducted between GIB and LS ($G_{0\_L}/L_{0\_L}$, $G_{0\_R}/L_{0\_R}$, $G_{200\_L}/L_{200\_L}$, and $G_{200\_R}/L_{200\_R}$) or between 0 and 200 mM NaCl treatments ($G_{200\_L}/G_{0\_L}$, $G_{200\_R}/G_{0\_R}$, $L_{200\_L}/L_{0\_L}$, and $L_{200\_R}/L_{0\_R}$). The results showed that 1,947 SSR-containing unigenes were differentially regulated in eight comparisons (S3 Table). Of which, 430, 655, 407, 152, 219, 409, 311, and 172 DEUs were identified in $G_{200\_L}/G_{0\_L}$, $L_{200\_L}/L_{0\_L}$, $G_{0\_L}/L_{0\_L}$, $G_{200\_L}/L_{200\_L}$, $G_{200\_R}/G_{0\_R}$, $L_{200\_R}/L_{0\_R}$, $G_{0\_R}/L_{0\_R}$, and $G_{200\_R}/L_{200\_R}$, respectively (Figs 3A and 3B). The number of upregulated genes induced by NaCl treatment was more than that of downregulated genes in $G_{200\_L}/G_{0\_L}$, $L_{200\_L}/L_{0\_L}$, $G_{200\_R}/G_{0\_R}$ and $G_{200\_R}/L_{200\_R}$. Regardless for GIB or LS, more DEUs were identified in leaves than in the roots. Meanwhile, LS showed greater transcriptional changes than GIB either in leaves or roots. Furthermore, 136 and 70 DEUs were common NaCl-responsive genes in leaves and

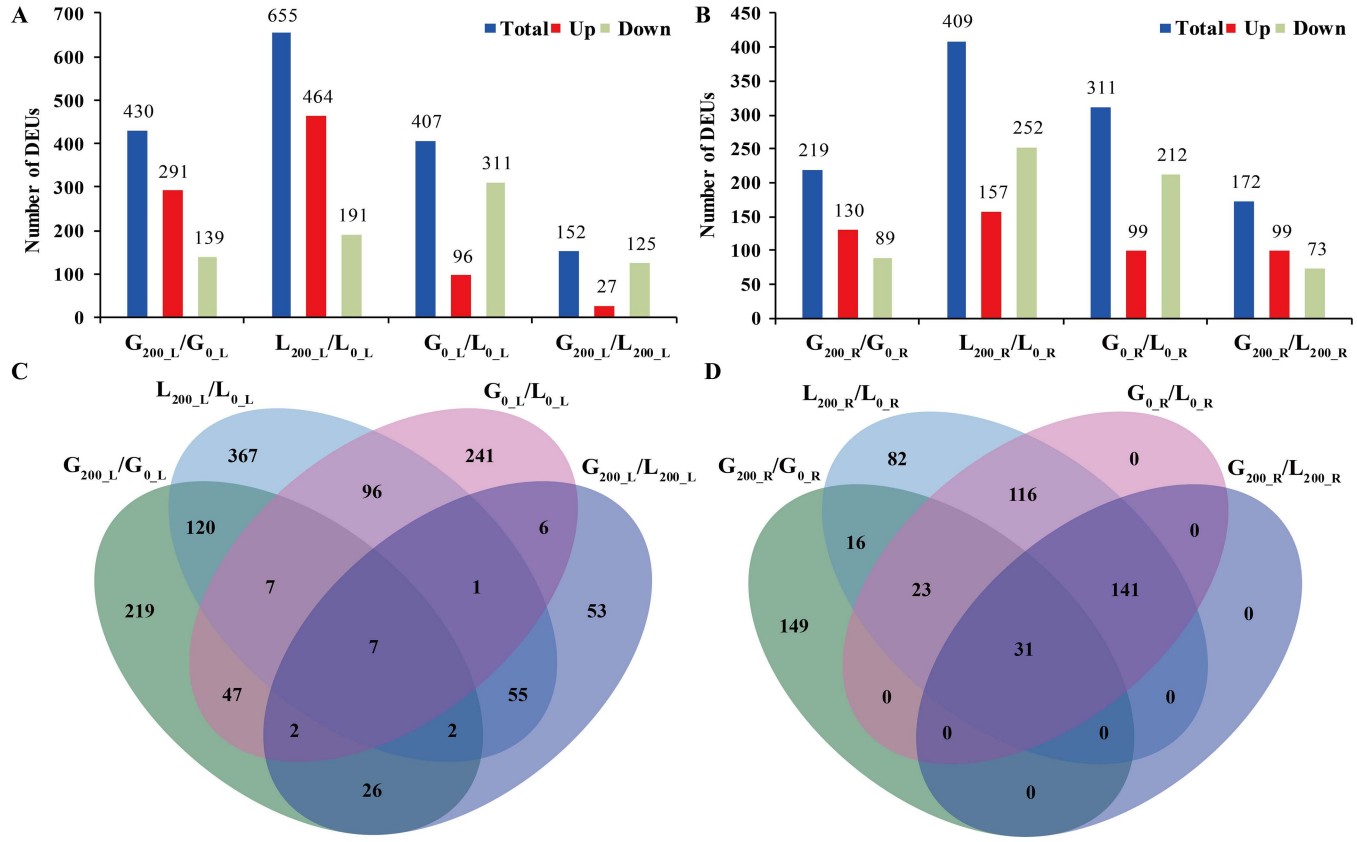

**Fig 3. Analysis of DEUs in two alfalfa varieties under salinity treatment. (a)** Number of DEUs in leaf. **(b)** Number of DEUs in root. **(c)** Venn diagrams of DEUs in leaf. **(d)** Venn diagrams of DEUs in root.

roots of the comparison groups, respectively. A total of 16 DEUs in leaves and 172 DEUs in roots were jointly regulated in $G_0/L_0$ and $G_{200}/L_{200}$. Moreover, 294 and 149 specific NaCl-responsive genes were detected in the GIB leaves and roots, respectively (Figs 3C and 3D).

## Functional classification of the SSR-containing DEUs by gene ontology

Gene ontology (GO) annotation results of NaCl-responsive DEUs between the two varieties were significantly different either in leaves or roots. Among 1,947 DEUs, 330 (116, 196, 91, and 24 unigenes in $G_{200\_L}/G_{0\_L}$, $L_{200\_L}/L_{0\_L}$, $G_{0\_L}/L_{0\_L}$, and $G_{200\_L}/L_{200\_L}$, respectively) and 62 (43, 23, 14, and 12 unigenes in $G_{200\_R}/G_{0\_R}$, $L_{200\_R}/L_{0\_R}$, $G_{0\_R}/L_{0\_R}$, and $G_{200\_R}/L_{200\_R}$, respectively) DEUs were assigned to 441 and 173 GO terms in leaves (S4A Table) and roots (S4B Table), respectively. A total of 132, 61, 43, and 39 GO terms were obtained from $G_{200\_R}/G_{0\_R}$, $R_{200\_R}/R_{0\_R}$, $G_{0\_R}/R_{0\_R}$, and $G_{200\_R}/R_{200\_R}$, respectively (S4B Table and Fig 4B), while this value reached 248, 286, 152, and 66 GO terms in $G_{200\_L}/G_{0\_L}$, $L_{200\_L}/L_{0\_L}$, $G_{0\_L}/L_{0\_L}$, and $G_{200\_L}/L_{200\_L}$, respectively (S4A Table and Fig 4A). The results of GO enrichment analysis ($P$-value ≤ 0.05) are listed in S5 Table. For GIB, the most enriched GO terms of NaCl-responsive DEUs included 17 (leaves) and 29 (roots) subcategories related to biological processes (e.g., fructose 6-phosphate metabolic process and brassinosteroid homeostasis), 14 (leaves) and 4 (roots) subcategories linked to cellular component (e.g., nucleotide-activated protein kinase complex, and anchored component of plasma membrane), and 17 (leaves) and 27 (roots) subcategories associated with molecular function(e.g., single-stranded telomeric DNA binding and DNA-directed DNA polymerase activity). NaCl -responsive DEUs of LS were

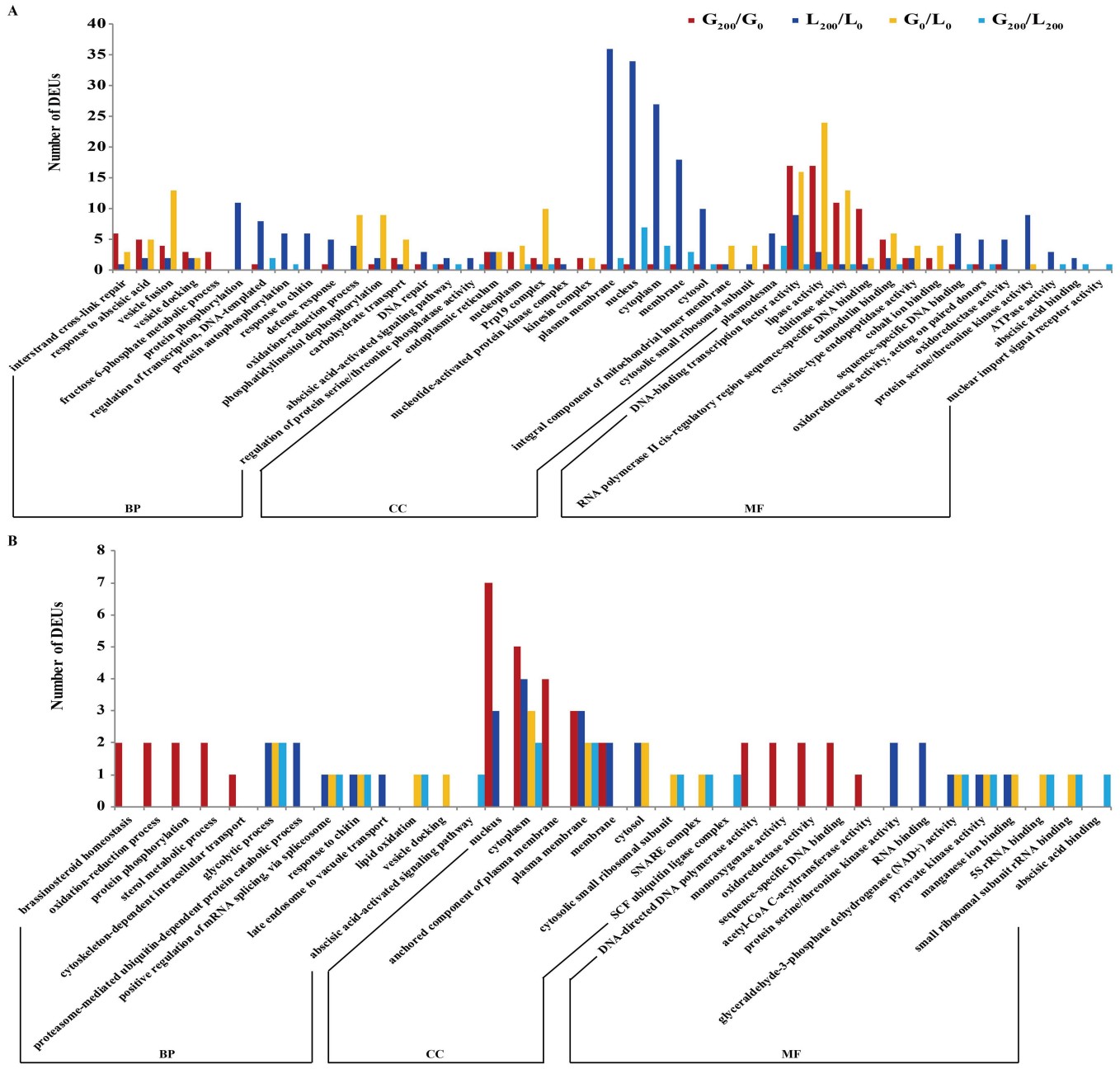

**Fig 4. Gene Ontology classification of DEUs in leaves. (a) and roots (b).** The top 5 abundant GO terms in BP (biological process), CC (cellular component) and MF (molecular function) for $G_{200}/G_0$, $L_{200}/L_0$, $G_0/L_0$ and $G_{200}/L_{200}$.

assigned to biological processes (e.g., response to stress and glycolytic process), cellular component (e.g., plasma membrane and integral component of membrane), and molecular function (e.g., DNA-binding transcription factor activity and manganese ion binding) (S5 Table). For DEUs between GIB and LS, 8 (leaves) and 10 (roots) GO terms of biological process (e.g., potassium ion transport, defense response to fungus, glycolytic process, and abscisic acid-activated signaling pathway), 8 (leaves) and 2 (roots) GO terms of cellular component (e.g., plasmodesma, nuclear chromosome,

spliceosomal complex, and SNARE complex), and 9 (leaves) and 10 (roots) GO terms of molecular function (e.g., proteasome binding, potassium ion transmembrane transporter activity, pyruvate kinase activity, and abscisic acid binding) were most enriched under NaCl stress (S5 Table).

## KEGG -based analysis of metabolic pathways associated with SSR-containing DEUs

A total of 75 ($G_{200\_L}/G_{0\_L}$),105 ($L_{200\_L}/L_{0\_L}$), 61 ($G_{0\_L}/L_{0\_L}$), 13 ($G_{200\_L}/L_{200\_L}$), 29 ($G_{200\_R}/G_{0\_R}$), 19 ($L_{200\_R}/L_{0\_R}$), 10 ($G_{0\_R}/L_{0\_R}$), and 7 ($G_{200\_R}/L_{200\_R}$) DEUs were respectively assigned to 76, 83, 50, 18, 47, 26, 17, and 12 pathways, which were classified into five categories: metabolism, genetic information processing, environmental information processing, cellular processes, and organismal systems (S6 Table). Among which, 15, 28, 16, 7, 12, 6, 4, and 6 significantly enriched pathways (*P*-value ≤ 0.05) were identified in $G_{200\_L}/G_{0\_L}$, $L_{200\_L}/L_{0\_L}$, $G_{0\_L}/L_{0\_L}$, $G_{200\_L}/L_{200\_L}$, $G_{200\_R}/G_{0\_R}$, $L_{200\_R}/L_{0\_R}$, $G_{0\_R}/L_{0\_R}$, and $G_{200\_R}/L_{200\_R}$, respectively (Table 3). For NaCl-responsive DEUs, the top two significantly enriched pathways were brassinosteroid biosynthesis and biosynthesis of secondary metabolites in $G_{200\_R}/G_{0\_R}$, proteasome and biosynthesis of amino acids in $L_{200\_R}/L_{0\_R}$, biosynthesis of secondary metabolites and flavonoid biosynthesis in $G_{200\_L}/G_{0\_L}$, and biosynthesis of secondary metabolites and MAPK signaling pathway in $L_{200\_L}/L_{0\_L}$. For DEUs between GIB and LS, mismatch repair, DNA replication, nucleotide excision repair, and homologous recombination were the pathways commonly enriched in $G_{0\_L}/L_{0\_L}$ and $G_{200\_L}/L_{200\_L}$. Glycolysis/gluconeogenesis, biosynthesis of amino acids, and linoleic acid metabolism were the pathways commonly enriched in $G_{0\_R}/L_{0\_R}$ and $G_{200\_R}/L_{200\_R}$. Furthermore, the pathways of glycolysis/gluconeogenesis, biosynthesis of amino acids, and carbon metabolism had the most significant enrichment in $G_{200\_R}/L_{200\_R}$. Meanwhile, cysteine and methionine metabolism, mismatch repair, DNA replication, nucleotide excision repair, and homologous recombination were the pathways enriched in $G_{200\_L}/L_{200\_L}$.

## SSR-containing DEUs related to salinity tolerance in alfalfa

Signal transductions, ion transport, metabolite biosynthesis, ROS regulation, and transcriptional regulation are important processes for plants to combat salinity stress [27,28]. Based on GO and KEGG pathway analyses, 34 DEUs encoding ion transport-related proteins, 29 DEUs related to metabolite biosynthesis, 20 DEUs associated with ROS regulation, 37 DEUs linked to signaling and kinase, and 68 DEUs mapped to different transcription factor families were identified in this study (S7A Table and Fig 5A). These DEUs were divided into three distinct expression patterns: (I) 139 salinity-responsive DEUs, (II) 36 salinity-responsive DEUs that were differentially expressed between GIB and LS, and (III) 13 DEUs differentially expressed between GIB and LS (S7B Table).

   The distribution of SSR motifs within the gene regions (exons, 3'UTR, and 5'UTR) of the five functional classifications was analyzed (Fig 5B). The locations of the largest number of SSR motifs in three regions were transcriptional regulation genes (6 exons, 18 3'UTR, and 17 5'UTR), followed by signaling and kinase (2) and ROS regulation (2) genes located in the exonic region, signaling and kinase (9) and metabolite biosynthesis (9) genes in the 5'UTR region, and ion transport (10) genes in the 5'UTR region (Fig 5B).

## Development of SSR markers linked to salinity tolerance genes in alfalfa

Combined with the SSR loci data from transcriptome sequencing, the SSR loci information of the 188 previously identified salinity tolerance-related unigenes were screened out and extracted. Among them, some unigenes contained two or more SSR loci; a total of 211 SSR primer pairs designed based on standard PCR primer criteria were synthesized for validation across four alfalfa varieties exhibiting contrasting salinity tolerance: GN3, GN5, GIB, and LS (S8 Table). A total of 200 out of 211 SSR loci were successfully localized on the 30 alfalfa chromosomes (Fig 6). Among 211 primer pairs, 110 primer pairs were detected with amplification products, of which 7 primer pairs (SSR121, SSR122, SSR125, SSR127, SSR165, SSR191, and SSR231) had relatively clear amplified bands. Their amplified bands differed in GN3, GN5, GIB, and LS (Fig 7), which could be used to distinguish different varieties. By searching the alfalfa genome annotation file (GTF/GFF)

**Table 3. Significantly enriched KEGG pathways for the DEUs ($P\text{-value} \leq 0.05$).**

| Pathway term | DEUs tested | | | | P-value | | | | Pathway ID |
|---|---|---|---|---|---|---|---|---|---|
| | $G_{200}/G_0$ | $L_{200}/L_0$ | $G_0/L_0$ | $G_{200}/L_{200}$ | $G_{200}/G_0$ | $L_{200}/L_0$ | $G_0/L_0$ | $G_{200}/L_{200}$ | |
| **Significantly enriched KEGG pathways for the DEUs in leaves** | | | | | | | | | |
| Metabolic pathways | 37 | 55 | 20 | 6 | 0.0013 | 0.0003 | 0.4355 | 0.6460 | mtr01100 |
| Biosynthesis of secondary metabolites | 32 | 39 | 13 | 3 | 0.0000 | 0.0000 | 0.1421 | 0.6249 | mtr01110 |
| Carbon metabolism | 3 | 7 | 1 | – | 0.3559 | 0.0495 | 0.8423 | – | mtr01200 |
| 2-Oxocarboxylic acid metabolism | 1 | 4 | 2 | – | 0.3551 | 0.0055 | 0.0575 | – | mtr01210 |
| Ascorbate and aldarate metabolism | 2 | 3 | 3 | – | 0.0731 | 0.0324 | 0.0073 | – | mtr00053 |
| Nitrogen metabolism | – | 3 | – | – | – | 0.0118 | – | – | mtr00910 |
| Glycerophospholipid metabolism | – | 3 | 3 | – | – | 0.1440 | 0.0389 | – | mtr00564 |
| Fatty acid degradation | 2 | 4 | – | – | 0.0772 | 0.0061 | – | – | mtr00071 |
| Glycerolipid metabolism | 1 | 4 | 1 | – | 0.5234 | 0.0299 | 0.4758 | – | mtr00561 |
| alpha-Linolenic acid metabolism | 4 | 4 | 1 | – | 0.0021 | 0.0095 | 0.3632 | – | mtr00592 |
| Sphingolipid metabolism | – | 1 | 2 | – | – | 0.3212 | 0.0213 | – | mtr00600 |
| Biosynthesis of unsaturated fatty acids | 2 | 1 | – | – | 0.0183 | 0.2665 | – | – | mtr01040 |
| Cysteine and methionine metabolism | 3 | 7 | 2 | 2 | 0.0624 | 0.0006 | 0.1839 | 0.0304 | mtr00270 |
| Arginine and proline metabolism | 1 | 4 | 3 | – | 0.4495 | 0.0152 | 0.0164 | – | mtr00330 |
| Phenylalanine metabolism | 2 | – | 1 | – | 0.0518 | – | 0.2690 | – | mtr00360 |
| Arginine biosynthesis | – | 3 | – | – | – | 0.0126 | – | – | mtr00220 |
| Alanine, aspartate and glutamate metabolism | – | 4 | – | – | – | 0.0058 | – | – | mtr00250 |
| Valine, leucine and isoleucine biosynthesis | 1 | 3 | 2 | – | 0.1277 | 0.0015 | 0.0069 | – | mtr00290 |
| Valine, leucine and isoleucine degradation | 3 | 5 | 2 | – | 0.0080 | 0.0005 | 0.0480 | – | mtr00280 |
| Lysine degradation | 1 | 3 | – | – | 0.3458 | 0.0298 | – | – | mtr00310 |
| Histidine metabolism | 2 | 2 | – | – | 0.0247 | 0.0538 | – | – | mtr00340 |
| Cyanoamino acid metabolism | 1 | 4 | – | – | 0.4766 | 0.0196 | – | – | mtr00460 |
| beta-Alanine metabolism | 1 | 3 | 1 | – | 0.3315 | 0.0262 | 0.2960 | – | mtr00410 |
| Selenocompound metabolism | – | 1 | – | 1 | – | 0.2074 | – | 0.0446 | mtr00450 |
| Taurine and hypotaurine metabolism | – | – | – | 1 | – | – | – | 0.0362 | mtr00430 |
| Pantothenate and CoA biosynthesis | 1 | 2 | 2 | – | 0.2606 | 0.0812 | 0.0297 | – | mtr00770 |
| Nicotinate and nicotinamide metabolism | – | 2 | – | – | – | 0.0284 | – | – | mtr00760 |
| Thiamine metabolism | 2 | – | – | – | 0.0148 | – | – | – | mtr00730 |
| Carotenoid biosynthesis | – | – | 2 | – | – | – | 0.0284 | – | mtr00906 |
| Terpenoid backbone biosynthesis | 3 | 1 | – | – | 0.0105 | 0.4910 | – | – | mtr00900 |
| Brassinosteroid biosynthesis | 2 | 1 | – | – | 0.0065 | 0.1623 | – | – | mtr00905 |
| Limonene and pinene degradation | 1 | 2 | – | – | 0.0760 | 0.0074 | – | – | mtr00903 |
| Phenylpropanoid biosynthesis | 6 | 5 | – | – | 0.0217 | 0.2345 | – | – | mtr00940 |
| Flavonoid biosynthesis | 5 | 6 | – | – | 0.0010 | 0.0012 | – | – | mtr00941 |
| Isoflavonoid biosynthesis | 3 | 4 | 1 | 1 | 0.0010 | 0.0003 | 0.1503 | 0.0549 | mtr00943 |
| Protein processing in endoplasmic reticulum | 3 | 4 | 6 | – | 0.4194 | 0.4925 | 0.0186 | – | mtr04141 |
| SNARE interactions in vesicular transport | – | 3 | 3 | – | – | 0.0159 | 0.0034 | – | mtr04130 |
| Homologous recombination | 2 | 2 | 5 | 2 | 0.1813 | 0.3336 | 0.0007 | 0.0232 | mtr03440 |
| Nucleotide excision repair | 2 | 2 | 5 | 2 | 0.1585 | 0.2973 | 0.0005 | 0.0197 | mtr03420 |
| DNA replication | 2 | 2 | 5 | 2 | 0.1266 | 0.2444 | 0.0002 | 0.0151 | mtr03030 |
| Mismatch repair | 2 | 2 | 5 | 2 | 0.1033 | 0.2039 | 0.0001 | 0.0120 | mtr03430 |
| Plant hormone signal transduction | 7 | 10 | 4 | – | 0.0300 | 0.0176 | 0.2534 | – | mtr04075 |
| MAPK signaling pathway – plant | 6 | 11 | 12 | 2 | 0.0032 | 0.0000 | 0.0000 | 0.0667 | mtr04016 |

*(Continued)*

**Table 3.** (Continued)

| Pathway term | DEUs tested | | | | P-value | | | | Pathway ID |
|---|---|---|---|---|---|---|---|---|---|
| | $G_{200}/G_0$ | $L_{200}/L_0$ | $G_0/L_0$ | $G_{200}/L_{200}$ | $G_{200}/G_0$ | $L_{200}/L_0$ | $G_0/L_0$ | $G_{200}/L_{200}$ | |
| Plant-pathogen interaction | 4 | 5 | 7 | 1 | 0.1094 | 0.1500 | 0.0012 | 0.4197 | mtr04626 |
| Circadian rhythm – plant | 2 | 4 | – | – | 0.0731 | 0.0055 | – | – | mtr04712 |
| **Significantly enriched KEGG pathways for the DEUs in roots** | | | | | | | | | |
| Biosynthesis of secondary metabolites | 12 | 5 | 3 | 2 | 0.0028 | 0.1231 | 0.2944 | 0.3473 | mtr01110 |
| Carbon metabolism | 1 | 2 | 2 | 2 | 0.6058 | 0.0969 | 0.0553 | 0.0250 | mtr01200 |
| Biosynthesis of amino acids | 3 | 3 | 2 | 2 | 0.0459 | 0.0104 | 0.0415 | 0.0185 | mtr01230 |
| Glycolysis/ Gluconeogenesis | – | 2 | 2 | 2 | – | 0.0280 | 0.0152 | 0.0066 | mtr00010 |
| Glycerophospholipid metabolism | – | 2 | 1 | – | – | 0.0188 | 0.1396 | – | mtr00564 |
| alpha-Linolenic acid metabolism | 2 | – | – | – | 0.0224 | – | – | – | mtr00592 |
| Linoleic acid metabolism | – | 1 | 1 | 1 | – | 0.0637 | 0.0464 | 0.0302 | mtr00591 |
| Purine metabolism | 2 | 2 | 2 | 1 | 0.0529 | 0.0185 | 0.0100 | 0.0919 | mtr00230 |
| Valine, leucine and isoleucine degradation | 2 | – | – | – | 0.0136 | – | – | – | mtr00280 |
| Thiamine metabolism | 2 | – | – | – | 0.0031 | – | – | – | mtr00730 |
| Terpenoid backbone biosynthesis | 2 | – | – | – | 0.0165 | – | – | – | mtr00900 |
| Brassinosteroid biosynthesis | 2 | – | – | – | 0.0013 | – | – | – | mtr00905 |
| Ribosome | 1 | 3 | 2 | 2 | 0.7545 | 0.0451 | 0.1113 | 0.0525 | mtr03010 |
| SNARE interactions in vesicular transport | – | 1 | 1 | 1 | – | 0.0785 | 0.0574 | 0.0375 | mtr04130 |
| Proteasome | 1 | 2 | – | – | 0.1908 | 0.0066 | – | – | mtr03050 |
| Homologous recombination | 2 | 1 | 1 | – | 0.0459 | 0.1733 | 0.1285 | – | mtr03440 |
| Nucleotide excision repair | 2 | 1 | 1 | – | 0.0392 | 0.1600 | 0.1183 | – | mtr03420 |
| DNA replication | 2 | 1 | 1 | – | 0.0304 | 0.1403 | 0.1035 | – | mtr03030 |
| Mismatch repair | 2 | 1 | 1 | – | 0.0242 | 0.1249 | 0.0919 | – | mtr03430 |

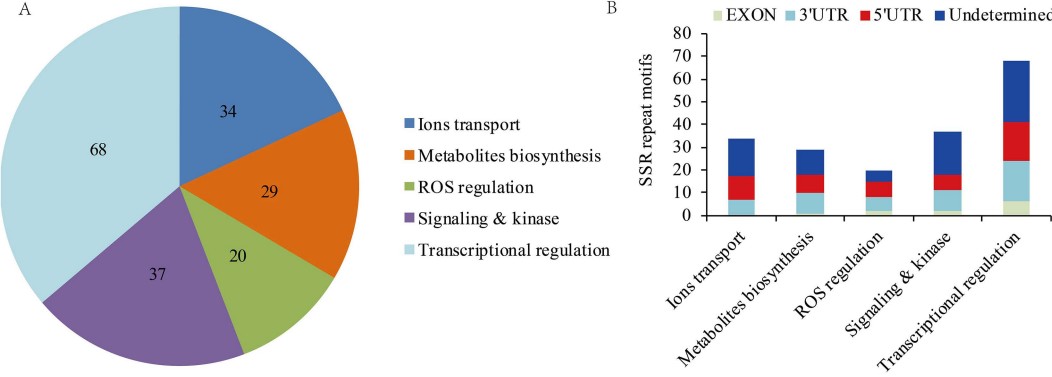

**Fig 5. Functional classification of SSR containing DEUs related to salinity stress tolerance (a) and location of SSRs in different regions of the salinity stress tolerance genes in each functional class (b).**

(S8 Table), we found that SSR122 and SSR125 loci were located in the 3'UTR region of the protein phosphatase 2C 12 (*PP2C12*, chr3.4) and *PP2C25* (chr5.4) gene, respectively; SSR165 was located in the 5'UTR region of the ethylene-responsive transcription factor (*ERF026*, chr8.4) gene; SSR121, SSR127, SSR191, and SSR231 were located in the undetermined regions of the leucine-rich repeat receptor-like serine/threonine-protein kinase (*LRR-RLK*, chr2.4), *PP2C2*

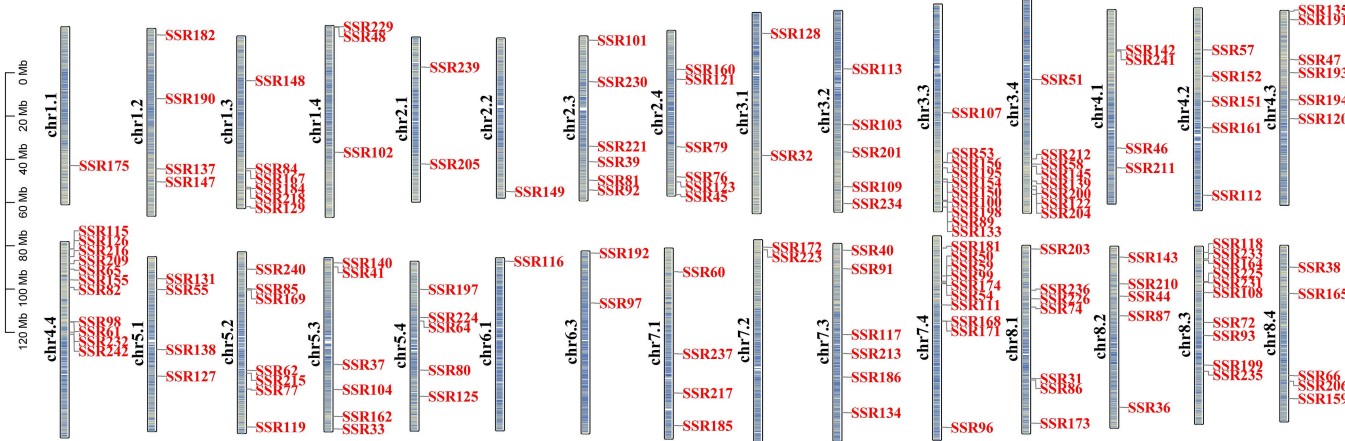

**Fig 6. Genomic localization of SSR markers on 30 *M. sativa* chromosomes.**

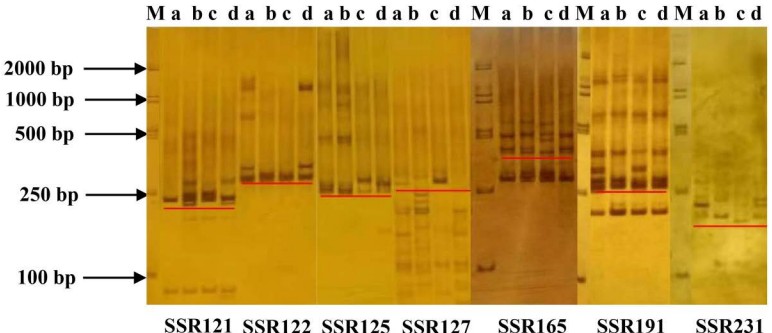

**Fig 7. Amplification of seven SSR primer pairs using genomic DNA of four alfalfa varieties as template.** The four bars **(a–d)** as a group, and the sequence of each group is GN3, GIB, LS and GN5.

(chr5.1), heat shock 70 kDa protein (*HSP70*, chr4.3), and two-component response regulator (*ARR5*, chr8.3) gene, respectively (Table 4).

### Genetic diversity and cluster analysis of four alfalfa varieties with contrasting salinity tolerance based on six polymorphic SSR markers

Among the seven SSR markers previously tested, except for SSR127, the six other polymorphic markers were used to assess the genetic diversity of GN3, GN5, GIB, and LS. A total of 31 bands were generated by the amplification, and 50 alleles were detected from six SSR markers (mean = 1.6667 alleles/marker) (S9 Table). Among them, SSR125 and SSR165 amplified the largest observed number of alleles (Na). The number of effective alleles (Ne) ranged from 1.0000 to 2.0000, with an average value of 1.3780. The average value of Nei's gene diversity (H, range of 0.0000–0.5000) and Shannon's information index (I, range of 0.0000–0.6931) were 0.2326 and 0.3526, respectively (S9 Table). The number of PIC ranged from 0.640 to 0.807 (mean = 0.7225) (S9 Table).

Based on the genetic distance coefficient, Unweighted Pair Group Method with Arithmetic Mean (UPGMA) cluster analysis of six polymorphic SSR markers data showed that the four different alfalfa varieties could be clustered into three main groups at a genetic distance coefficient of 0.31. The first group comprised LS (high salinity sensitivity), the second group included GIB and GN5 (high salinity tolerance), and the third group consisted of GN3 (salinity sensitivity) (Fig 8).

**Table 4. Development the salinity stress tolerance related seven SSR markers.**

| Primer ID | Gene ID | Annotation | Gene name | (Motif) repeats | Amplified product (bp) | Position | Chr. |
|---|---|---|---|---|---|---|---|
| SSR121 | Cluster-27126.42501 | Leucine-rich repeat receptor-like serine/threonine-protein kinase | *LRR-RLK/ At3g14840* | (T)14 | 230–270 | undetermined | chr2.4 |
| SSR122 | Cluster-27126.45536 | Protein phosphatase 2C 12, | *PP2C12* | (A)10 | 270–280 | 3'UTR | chr3.4 |
| SSR125 | Cluster-27126.43113 | Protein phosphatase 2C 25 | *PP2C25* | (T)11 | 249–260 | 3'UTR | chr5.4 |
| SSR127 | Cluster-27126.43458 | Protein phosphatase 2C 2 | *PP2C2* | (A)11,(A)15, (TTG)6 | 250–260 | undetermined | chr5.1 |
| SSR165 | Cluster-27126.29621 | Ethylene-responsive transcription factor ERF026 | *ERF026* | (CTT)5 | 280–400 | 5'UTR | chr8.4 |
| SSR191 | Cluster-27126.57746 | Heat shock 70 kDa protein | *HSP70* | (AG)10 | 241–300 | undetermined | Chr4.3 |
| SSR231 | Cluster-27126.24781 | Two-component response regulator ARR5 | *ARR5* | (AAT)9 | 189–230 | undetermined | Chr8.3 |

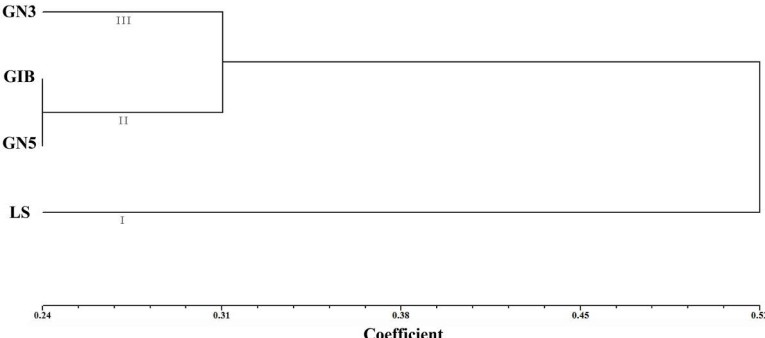

**Fig 8. Cluster analyses of four alfalfa varieties differing in salinity tolerance.**

## Discussion

Salinity stress is a key environmental factor limiting the growth of plants worldwide. Alfalfa, as a vital forage crop, is relatively susceptible to salinity stress [29]. Notably, a great variability in salinity tolerance exists among alfalfa varieties [22,30,31]. However, accurate identification and differentiation between varieties is limited due to the self-incompatibility and cross-pollination behavior of autotetraploid species [32]. SSR loci are distributed throughout plant genomes and have been effectively used to analyze genetic diversity [16] and identify genotypes [17,33] in tetraploid alfalfa. Although many SSR markers have been developed in alfalfa, the development of salinity tolerance-associated SSR markers in alfalfa remains rarely reported.

### Transcriptome sequencing as an effective method for developing SSR markers in alfalfa

Transcriptome sequencing has been effectively and rapidly utilized to identify key stress-tolerant genes [10,34], and develop SSR markers in many plant species, such as *Anoectochilus roxburghii* [35], *Miscanthus* [15], and *Chimonanthus praecox* [36]. Nie et al. detected 16,566 gene-associated SSR loci from the transcriptomic sequences of *Miscanthus* under drought stress, which promoted the development of SSR markers related to drought resistance [15]. Liu et al. identified 64,440 SSRs through transcriptome sequencing of *C. praecox*, and 75 polymorphic SSR markers were used to analyze the genetic relationships of 12 *C. praecox* varieties [36]. In this study, 129,563 unigenes were obtained through transcriptome sequencing on the roots and leaves of GIB and LS treated with salinity stress (Table 1). Using the MISA

script, 38,370 SSR loci distributed among 28,039 unigenes were detected, with a distribution frequency of 21.64% (Table 1). This frequency was lower than that reported for *Polygonatum odoratum* (29.47%) [14] but higher than that of *Anoectochilus emeiensis* (11.18%) [37]. Among the SSR repeat units, mononucleotide (67.32%) repeats were the most abundant, followed by trinucleotide (15.61%), and dinucleotide (14.53%) repeat units (Table 2). This finding was similar to the sequencing results of *P. odoratum* [14] and *Chrysanthemum* [10] but different from those of *Miscanthus* [15] and *C. praecox* [36]. In our study, A/T (67.07%) was the most abundant mononucleotide repeat (Fig 1), which was similar to *Chrysanthemum* [10]. Meanwhile, we found that AG/CT (7.28%) repeats were the dominant dinucleotide motifs (Fig 1), which was also consistent with those reported for other species such as *Miscanthus* [15] and *C. praecox* [36]. In trinucleotide repeats, AAG/CTT (4.11%) was the more frequent (Fig 1), which was consistent with those reported for some dicotyledonous plants such as *C. praecox* [36] and *Carex rigescens* [38]. By contrast, this finding was different from those reported for some monocotyledons plants such as *Miscanthus* [15] and *Dendrocalamus brandisii* [39]. These differences may be due to methodological variations, differences in database availability, and species-specific characteristics.

A total of 23,159 alternate primer pairs of SSRs were designed based on the transcriptome data in this study (S1 Table). The amplification efficiency of randomly selected primer pairs was 81.8%, which was higher than that in *P. odoratum* (60.2%) [14] and *Miscanthus* (75.3%) [15]. The abovementioned results indicated that the SSR primers developed based on transcriptome sequencing in this study are effective and feasible.

## Development and application of SSR markers associated with salinity stress tolerance in alfalfa

Functional markers are derived from polymorphic sites within genes or regulatory sequences that are directly linked to phenotypic trait variation [8]. SSRs identified in transcriptome data are located in coding and non-coding regions and often correlate with specific functions; thus, mining SSR markers from transcriptome data realize targeted identification of trait-associated loci [10,40]. For example, Shi et al. developed eight polymorphic SSR markers related to flower colors based on transcriptomic sequences of genes involved in carotenoid or anthocyanin synthesis [10]. To identify SSR markers associated with salinity stress tolerance in alfalfa, 1,947 SSR-containing DEUs were identified in pairwise comparisons (S3 Table). GO and KEGG pathway annotation showed that the biosynthesis of secondary metabolites was most enriched in $G_{200\_L}/G_{0\_L}$ and $L_{200\_L}/L_{0\_L}$ (Table 3). Among the 1,947 DEUs, 188 SSR-containing DEUs were identified to be related to ion transport (34), metabolite biosynthesis (29), ROS regulation (20), signalling pathway (37), and transcription regulation (68) (S7A Table and Fig 5A). These processes are known to regulate plant salinity tolerance [27,28], which indicates that they may contribute to the differences in salinity tolerance between the two alfalfa varieties.

To validate whether the abovementioned results can be used as SSR markers to identify alfalfa varieties with different salinity tolerance, the SSR loci information of the 188 previously identified salinity-responsive related DEUs were screened out and extracted, and 211 SSR primer pairs (S8 Table) were synthesized for validation in GN5, GIB, GN3, and LS [5,22]. A total of seven markers produced clear amplification bands, with six (SSR121, SSR122, SSR125, SSR165, SSR191 and SSR231) showing polymorphism (Fig 7). Therefore, they could be used to distinguish different varieties. For instance, SSR231, SSR121, and SSR122, SSR121 and SSR125, SSR122 and SSR165, and SSR122 and SSR191 can distinguish four different alfalfa varieties with 100% accuracy. SSR127 and SSR165 can distinguish LS from the other alfalfa varieties (Fig 7). Interestingly, these SSRs originated from the transcript sequences of *LRR-RLK*, *PP2C12*, *PP2C25*, *ERF026*, *HSP70*, and *ARR5* genes (Table 4). *LRR-RLKs*, *PP2Cs*, *ERFs*, and *HSPs* are important regulators of plant responses to abiotic stress. Transcriptome sequencing results showed that *PP2C12*, *PP2C25*, *ERF026* in roots, and *ARR5* in leaves were upregulated by NaCl treatment in GIB, while in LS, they were unchanged. Meanwhile, *LRR-RLK* was downregulated by NaCl treatment in LS, while in GIB, it was unchanged, *and HSP70* was upregulated by NaCl treatment in both varieties (S3 Table). Overexpression of *OsSTLK* (a member of LRR-RLK) [41], *PnLRR-RLK27* [42], *PagERF021* [43], and *MdHSP70–38* [44] enhanced the salinity stress tolerance of transgenic plants through regulating ROS scavenging system. Overexpression of *AtARR22* [45] and *AtARR5* [46] genes in *Arabidopsis* enhanced the

freezing tolerance of transgenic lines by improving cell membrane integrity. *PP2CA* together with *ABI1* inhibited SnRK2.4 activity and regulated plant responses to salinity [47], and overexpression of *BpPP2C1* in *Betula platyphylla* improved the salinity tolerance of transgenic lines, and knockout of *BpPP2C1* exhibited sensitive to salt stress [48]. Therefore, we speculate that these genes may serve as marker genes for distinguishing the four alfalfa varieties with contrasting salinity tolerance.

PIC values represent the informativeness of molecular marker [10]. Here, the PIC values of six SSRs ranged from 0.640 to 0.807, and the average PIC value was 0.7225 (PIC > 0.5) (S9 Table), which suggests that the newly developed SSR markers in this study could be further used for variety authentication and genetic analysis in alfalfa.

Meanwhile, polymorphic SSR markers have been widely used for classification in many plants, such as *Chrysanthemum* [10], chrysanthemum [13], and *Arctium lappa* [49]. In this study, through UPGMA clustering based on genetic distances, the six newly developed SSR markers classified four alfalfa varieties into three groups: group I (LS), group II (GIB and GN5), and group III (GN3) (Fig 8). This finding is similar to our previous results on salinity tolerance in alfalfa, which categorized the varieties into three groups: high salinity tolerance (GIB and GN5), salinity sensitivity (GN3), and high salinity sensitivity (LS) [5,22]. The combination of salt tolerance, PCR, and cluster analyses indicated a potential association between the six newly developed SSR markers and salinity tolerance in alfalfa.

## Conclusions

In this study, 38,370 SSR loci were detected from transcriptome sequencing, and 23,159 primer pairs of SSRs were successfully designed based on these loci. Furthermore, 188 SSR-containing DEUs were involved in ion transport, metabolite biosynthesis, ROS regulation, signaling pathway, and transcription regulation, which were all related to salinity tolerance. Six polymorphic SSR markers derived from transcriptome data were validated and found to be involved in signaling pathway and ROS regulation. Cluster analysis grouped the four alfalfa varieties into three groups: high salinity-sensitive variety LS (group I), high salinity-tolerant varieties GIB and GN5 (group II), and salinity-sensitive variety GN3 (group III). This classification pattern suggests a potential association between the six SSR markers and salinity tolerance in alfalfa. Whether these markers are functionally linked to salinity-tolerant genes and whether they can reliably identify salinity tolerance in alfalfa needs further verification. This study provides a solid foundation for the large-scale development of markers related to specific traits, genetic diversity studies, classification, and molecular-assisted breeding in alfalfa.

## Supporting information

**S1 Table. List of 11 SSR random primer pairs used for PCR amplification verification in GIB.**
(XLSX)

**S2 Table. List of 23,159 SSR primer pairs were designed in this study.**
(XLSX)

**S3 Table. List of DEUs in eight groups based on pairwise comparisons.**
(XLSX)

**S4A Table. The GO function classification of all DEUs in four comparisons of leaves.**
(XLSX)

**S4B Table. The GO function classification of all DEUs in four comparisons of roots.**
(XLSX)

**S5A Table. The enriched GO terms of all DEUs in four comparisons of leaves.**
(XLSX)

**S5B Table. The enriched GO terms of all DEUs in four comparisons of roots.**
(XLSX)

**S6A Table. Overview of all 104 KEGG pathways for DEUs in four comparisons of leaves.**
(XLSX)

**S6B Table. Overview of all 104 KEGG pathways for DEUs in four comparisons of roots.**
(XLSX)

**S7A Table. Listed functional classification of SSR-containing DEUs related to salinity stress tolerance.**
(XLSX)

**S7B Table. Listed classification of SSR-containing DEUs related to salinity stress tolerance based on their expression patterns.**
(XLSX)

**S8 Table. List of 211 SSR primer pairs used for the development of salinity-tolerant related SSR markers.**
(XLSX)

**S9 Table. Genetic diversity analysis with polymorphism SSR markers developed in this study.**
(XLSX)

## Author contributions

**Conceptualization:** Rugang Yu, Guoliang Wang, Xueling Du.

**Data curation:** Rugang Yu, Xin Chen.

**Formal analysis:** Rugang Yu, Xin Chen, Daniel Basigalup.

**Funding acquisition:** Guoliang Wang, Xueling Du.

**Investigation:** Rugang Yu, Xin Chen, Hui Zhang, Qiting Zhang, Xinyi Chen, Yanqiu Dong, Liwei Chen.

**Methodology:** Guoliang Wang, Xueling Du.

**Project administration:** Rugang Yu, Guoliang Wang, Xueling Du.

**Supervision:** Rugang Yu, Guoliang Wang, Xueling Du.

**Validation:** Xin Chen, Hui Zhang, Qiting Zhang, Xinyi Chen, Yanqiu Dong, Liwei Chen.

**Visualization:** Xin Chen.

**Writing – original draft:** Rugang Yu, Xin Chen.

**Writing – review & editing:** Hui Zhang, Daniel Basigalup, Guoliang Wang, Xueling Du.

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
