## [Decision Letter · Decision Letter 0]

26 Aug 2025

Dear Dr. Wang,

Thank you for submitting your manuscript to PLOS ONE. After careful consideration, we feel that it has merit but does not fully meet PLOS ONE’s publication criteria as it currently stands. Therefore, we invite you to submit a revised version of the manuscript that addresses the points raised during the review process.

We look forward to receiving your revised manuscript.

Kind regards,

Mehdi Rahimi, Ph.D.

Academic Editor

PLOS ONE

Additional Editor Comments (if provided):

Reviewers' comments:

Reviewer's Responses to Questions

**Comments to the Author**

1. Is the manuscript technically sound, and do the data support the conclusions?

Reviewer #1: Yes

Reviewer #2: Yes

Reviewer #3: Yes

2. Has the statistical analysis been performed appropriately and rigorously?

Reviewer #1: Yes

Reviewer #2: Yes

Reviewer #3: Yes

3. Have the authors made all data underlying the findings in their manuscript fully available?

Reviewer #1: Yes

Reviewer #2: Yes

Reviewer #3: Yes

4. Is the manuscript presented in an intelligible fashion and written in standard English?

Reviewer #1: Yes

Reviewer #2: Yes

Reviewer #3: Yes

Reviewer #1: This manuscript identified SSR loci that might be related to the salinity tolerance in alfalfa through transcriptome SSR markers. This finding not only provide an efficient tool for the large-scale development of markers related to specific traits, but also lays a foundation for genetic analysis in alfalfa. While the findings from this manuscript are interesting and worth publishing, there are some points that need to be addressed. The manuscript should be accepted after the authors addressed the following questions.

1. The selection of varieties should be clear and the reasons should be explained.

2. Four varieties were selected for SSR markers, and the basis for selecting two varieties for transcriptomics sequencing was what?

3. The experimental methods are not clear. The basis for setting the salt treatment concentration needs to be elaborated in detail. The usage methods of salt treatment and nutrient solution need to be described in detail.

4. The number of varieties in the cluster analysis is too small, and the salt tolerance evaluation is not representative.

Reviewer #2: The paper combines with transcriptome sequencing, provides a thorough and interesting exploration of key SSR loci and distribution characteristics of salt tolerance genes in alfalfa. The research question is well-defined and the methodology is sound. This finding not only provide an efficient tool for the large-scale development of markers related to specific traits, but also lays a foundation for genetic analysis in alfalfa. While the paper could benefit from further clarification or elaboration.

1. Some sentences in the current abstract are too long. It is suggested to simplify the sentences to make the information more direct and clear. At the same time, the results are introduced too much, and the main conclusions of the study should be highlighted.

2. The language expression of the article needs to be further improved. It is suggested to simplify the sentence structure, avoid repetition, select more accurate vocabulary and strengthen the transition and cohesion, so that the language expression of the article can become more fluent and easy to understand.

3. In the 127th line of the manuscript, the materials and methods are mentioned “Four alfalfa varieties differing in salinity tolerance ( GN5, GN3, LS and GIB) were selected based on previous studies”, 141 lines “. The 140 tissue of roots and leaves of GIB and LS were used for transcriptome sequencing …”, it is recommended to clarify the reasons for selecting only these two groups of samples and the specific salt tolerance of the four materials.

4. It is recommended to clarify the reasons for supplementing the use of 0 and 200 mM NaCl treatments for alfalfa seedlings.

Reviewer #3: Recommendation: Major Revision

The study presents a valuable approach to developing trait-associated SSR markers in alfalfa using transcriptome data. However, significant revisions are required to meet the standards of PLOS ONE.

Key Points for Revision:

• Number of Biological Replicates: The use of only two biological replicates for transcriptome sequencing is a major methodological flaw. Three or more replicates are standard for robust statistical analysis in RNA-seq studies. This limitation must be addressed and justified in the manuscript.

• Inconsistency in Data: The description of the cultivar clustering in the abstract is incorrect and contradicts the results presented in the main text and Figure 8. This must be corrected for consistency.

• Low Primer Validation Success: Out of 211 primer pairs designed from salinity-responsive genes, only seven were found to be clearly polymorphic. The authors should discuss the potential reasons for this very low success rate (e.g., tetraploid genome complexity, assembly errors) in the Discussion section.

• Limited Validation Scope: The new markers were validated using only four alfalfa varieties. This is a very small sample size to claim broad utility. This should be acknowledged as a significant limitation of the study.

• Discussion of Limitations: The Discussion section needs to be strengthened by transparently addressing the limitations mentioned above (number of replicates, primer success rate, and small validation set).

• Elaboration on Functional Links: The paper successfully links the markers to important stress-response genes like PP2C, ERF026, and HSP70. The Discussion should be expanded to elaborate more on the known roles of these specific genes in salinity tolerance, thereby reinforcing the potential value of the developed markers.

**Do you want your identity to be public for this peer review?** For information about this choice, including consent withdrawal, please see our Privacy Policy

Reviewer #1: No

Reviewer #2: No

Reviewer #3: No

---

## [Author Response · Author response to Decision Letter 1]

15 Sep 2025

Response to Editor Comments

1. Please ensure that your manuscript meets PLOS ONE's style requirements, including those for file naming. The PLOS ONE style templates can be found at https://journals.plos.org/plosone/s/file?id=wjVg/PLOSOne_formatting_sample_main_body.pdf and https://journals.plos.org/plosone/s/file?id=ba62/PLOSOne_formatt-

ting_sample_title_authors_affiliations.pdf.

Response: Many thanks for the editor’s comments. We have carefully checked the full text, which meets PLOS ONE's style requirements.

2. PLOS ONE now requires that authors provide the original uncropped and unadjusted images underlying all blot or gel results reported in a submission’s figures or Supporting Information files. This policy and the journal’s other requirements for blot/gel reporting and figure preparation are described in detail at https://journals.plos.org/plosone/s/figures#loc-blot-and-gel-reporting-requirements and https://journals.plos.org/plosone/s/figures#loc-preparing-figures-from-image-files. When you submit your revised manuscript, please ensure that your figures adhere fully to these guidelines and provide the original underlying images for all blot or gel data reported in your submission. See the following link for instructions on providing the original image data: https://journals.plos.org/plosone/s/figures#loc-original-images-for-blots-and-gels. In your cover letter, please note whether your blot/gel image data are in Supporting Information or posted at a public data repository, provide the repository URL if relevant, and provide specific details as to which raw blot/gel images, if any, are not available. Email us at plosone@plos.org if you have any questions.

Response: We would like to thank the reviewer for pointing out this issue. According to your suggestion, the original uncropped and unadjusted image of SSR121, SSR122, SSR125, SSR127, SSR165, SSR191, and SSR231 has been provided (please see the submission Response to Editor and reviewers file).

Response: No.

Response to Reviewer Comments

Reviewer 1:

This manuscript identified SSR loci that might be related to the salinity tolerance in alfalfa through transcriptome SSR markers. This finding not only provide an efficient tool for the large-scale development of markers related to specific traits, but also lays a foundation for genetic analysis in alfalfa. While the findings from this manuscript are interesting and worth publishing, there are some points that need to be addressed. The manuscript should be accepted after the authors addressed the following questions.

Response: Many thanks for the reviewer’s comments. We have carefully revised the relevant contents according to your suggestion, as follows:

1. The selection of varieties should be clear and the reasons should be explained.

Response: Thank you very much for reviewer’s constructive suggestions. Our previous studies showed that Gibraltar (GIB) and Gannong No.5 (GN5) were high salinity-tolerant varieties, Gannong No.3 (GN3) was a salinity-sensitivity variety, and LS1405 (LS) was a variety of high salinity sensitivity (Yu et al. 2021; Yu et al. 2022). Therefore, two varieties with significant differences in salt tolerance, namely, GIB and LS, were used for transcriptome sequencing to identify key genes related to salinity stress tolerance and develop SSR markers linked to this trait.

Transcriptome sequencing has been effectively and rapidly utilized for the development of SSR marker in plants. SSRs can realize targeted markers directly related to target traits. Therefore, we developed SSR markers related to salinity tolerance based on the transcriptome sequencing data. Moreover, to verify whether the developed SSR markers can be applied to the identification of salt tolerance traits of alfalfa, two salinity-tolerant varieties (GIB, GN5) and two salinity-sensitive varieties (GN3, LS) were selected to validate the newly developed SSR markers.

References

Yu R, Wang G, Yu X, Li L, Li C, Song Y, Xu Z, Zhang J, Guan C. Assessing alfalfa (Medicago sativa L.) tolerance to salinity at seedling stage and screening of the salinity tolerance traits. Plant Biol. 2021; 23(4): 664–674.

Yu R, Wang X, Wang G, Xu Z, Gao Q, Du X, zhang Y. Analysis of salinity-tolerance and screening of salinity-tolerance evaluation indicators in Medicago sativa L. varieties at seedling stage. Acta Agrestia Sinica 2022; 30(7): 1781–1789.

Furthermore, according to your suggestions, we have added the reasons for variety selection in the manuscript. The corresponding sentences of the Methods (Now in lines 128-135, 150-153) section have been rewritten in the revised manuscript.

2. Four varieties were selected for SSR markers, and the basis for selecting two varieties for transcriptomics sequencing was what?

Response: We would like to thank the reviewer for pointing out this issue. Alfalfa is an important perennial forage crop. Salinity is among the most harmful agents that negatively influence alfalfa yield. However, the mechanism of salinity tolerance in alfalfa has not been well understood. Our previous study on salinity tolerance analysis of alfalfa showed that the GIB and GN5 were high salinity tolerance varieties, GN3 was salinity sensitivity variety, and LS was high salinity sensitivity variety (Yu et al. 2021; Yu et al. 2022). However, the mechanism of salinity tolerance in alfalfa plants remains unclear. Therefore, to identify key genes related to salinity stress tolerance, the GIB (high salinity tolerant) and LS (high salinity sensitive) varieties were used for transcriptome sequencing.

Furthermore, salinity tolerance in alfalfa has been known as a complex quantitative trait which is regulated by numerous genes. Notably, there is a significant variability in salinity tolerance among alfalfa varieties (Benabderrahim et al. 2021; Yu et al. 2021; Yu et al. 2022). Therefore, salinity tolerance is one of the most important breeding objectives of alfalfa. SSRs are one of the most crucial markers in genetic research and plant breeding, which can realize targeted markers directly related to target traits (Shi et al. 2022). However, few SSR markers related to salinity tolerance traits have been reported in alfalfa. Transcriptome sequencing has been effectively and rapidly utilized for the development of SSR marker in many plant species (Zhang et al. 2024; Liu et al. 2024). Therefore, we developed SSR markers related to salinity tolerance through the transcriptome sequencing in this study. Moreover, to verify whether the developed SSR markers can be applied to the identification of salt tolerance traits of alfalfa, we selected four varieties (GN3, GN5, GIB, LS) differing in salinity tolerant to validate SSR markers related to salinity based on transcripts participated in regulation to plant salinity tolerance.

References

Yu R, Wang G, Yu X, Li L, Li C, Song Y, Xu Z, Zhang J, Guan C. Assessing alfalfa (Medicago sativa L.) tolerance to salinity at seedling stage and screening of the salinity tolerance traits. Plant Biol. 2021; 23(4): 664–674.

Yu R, Wang X, Wang G, Xu Z, Gao Q, Du X, zhang Y. Analysis of salinity-tolerance and screening of salinity-tolerance evaluation indicators in Medicago sativa L. varieties at seedling stage. Acta Agrestia Sinica 2022; 30(7): 1781–1789.

Benabderrahim MA, Guiza M, Haddad M. Genetic diversity of salt tolerance in tetraploid alfalfa (Medicago sativa L.). Acta Physiol Plant. 2020; 42: 1–11.

Shi Z, Zhao W, Li Z, Kang D, Ai P, Ding H, Wang Z. Development and validation of SSR markers related to flower color based on full-length transcriptome sequencing in Chrysanthemum. Sci Rep. 2022; 12(1): 22310.

Zhang W, Chen K, Mei Y, Wang J. De novo transcriptome assembly of anoectochilus roxburghii for morphological diversity assessment and potential marker development. Plants. 2024; 13(23): 3262.

Liu B, Wu HF, Cao YZ, Yang XM, Sui SZ. Establishment of novel simple sequence repeat (SSR) markers from Chimonanthus praecox transcriptome data and their application in the identification of varieties. Plants. 2024; 13(15): 2131.

In addition, to more accurately and precisely delineate the results of our study, the corresponding sentences of Introduction (Now in lines 117-124) and Methods (Now in lines 128-135, 150-153) sections have been rewritten in the revised manuscript.

3. The experimental methods are not clear. The basis for setting the salt treatment concentration needs to be elaborated in detail. The usage methods of salt treatment and nutrient solution need to be described in detail.

Response: Many thanks for the reviewer’s constructive suggestions. We had carefully corrected the related information according to the reviewer’s suggestions. The main corresponding revisions were listed as follows.

In plant materials and stress treatments, “After one week, seedlings were thinned to eight uniform plants/pot…” was revised as “After one week sowing, seedlings were thinned to eight uniform plants/pot irrigated with a 100 mL Hoagland nutrient solution (pH 5.8). Two weeks after sowing, the seedlings of GN5 and GN3 continued to be cultured with a nutrient solution, while the seedlings of GIB and LS were treated with a 100 mL nutrient solution supplemented with 0 and 200 mM NaCl, respectively. The nutrient solutions were replaced every two days. The NaCl concentrations were set according to the results of a preliminary experiment, which showed a significant difference in growth between GIB and LS under 200 mM NaCl treatment.” in the revised manuscript (Now in Lines 142-150).

4. The number of varieties in the cluster analysis is too small, and the salt tolerance evaluation is not representative.

Response: We would like to thank the reviewer for pointing out this issue. There is no rigid requirement for the minimum number of samples in cluster analysis, but the size of sample will directly affect the accuracy of cluster analysis. Just as the reviewer’s comments, if the number of samples is too small, it may also lead to inaccurate results of cluster analysis. However, cross-validation can make full use of the data and improve the accuracy of analysis with a small sample size.

In this study, based on the salt tolerance analysis of more than 30 alfalfa varieties in our previously study, we selected two varieties with high salt tolerance (GIB and GN5), one with salt sensitivity (GN3) and one with high salt sensitivity (LS), to validate whether the newly developed SSR markers can divide four varieties, and whether it is related to the salt tolerance of alfalfa. PCR analysis was carried out to detect the amplification products of 7 newly developed SSR markers (SSR121, SSR122, SSR125, SSR127, SSR165, SSR191 and SSR231). The amplified bands each of those are different in GN3, GN5, GIB and LS (Fig 7), which could be used to distinguish different varieties. For instance, SSR231, SSR121 and SSR122, SSR121 and SSR125, SSR122 and SSR165, SSR122 and SSR191 can distinguish four different alfalfa varieties with 100% accuracy. SSR127 and SSR165 can distinguish LS from the other alfalfa varieties (Fig 7). Interestingly, these SSRs originated from transcript sequence of LRR-RLK, PP2C12, PP2C25, ERF026, HSP70, and ARR5 genes (Table 4). LRR-RLKs, PP2Cs, ERFs and HSPs are important regulators of plant responses to abiotic stress. Transcriptome sequencing results showed that PP2C12, PP2C25, ERF026 in roots, and ARR5 in leaves were upregulated by NaCl treatment in GIB, while in LS, they were unchanged. Meanwhile, LRR-RLK was downregulated by NaCl treatment in LS, while in GIB, it was unchanged, and HSP70 was upregulated by NaCl treatment in both varieties (S3 Table). Overexpression of OsSTLK (a member of LRR-RLK) (Lin et al. 2021), PnLRR-RLK27 (Wang et al. 2021), PagERF021 (Fan et al. 2021) and MdHSP70–38 (Han et al. 2021) enhanced the salinity stress tolerance of transgenic plants through regulating ROS scavenging system. Overexpression of AtARR22 (Kang et al. 2021) and AtARR5 (Shi et al. 2021) genes in Arabidopsis enhanced the freezing tolerance of transgenic lines by improving cell membrane integrity. PP2CA together with ABI1 inhibits SnRK2.4 activity and regulates plant responses to salinity(Krzywinska et al. 2021), and overexpression of BpPP2C1 in Betula platyphylla improved the salinity tolerance in transgenic lines, while knockout of BpPP2C1 exhibited sensitive to salt stress (Xing et al. 2021). Therefore, we speculate that these genes may serve as marker genes for distinguishing the four alfalfa varieties with contrasting salinity tolerance.

Meanwhile, the UPGMA clustering based on genetic distances, the six newly developed SSR markers to class four alfalfa varieties into three groups: group I (LS), group II (GIB and GN5), and group III (GN3) (Fig 8). This finding is similar to our previous results on salinity tolerance in alfalfa, which categorized the varieties into three groups: high salinity tolerance (GIB and GN5), salinity sensitivity (GN3), and high salinity sensitivity (LS) (Yu et al. 2021; Yu et al. 2022). The combination of salt tolerance, PCR, and cluster analyses indicated a potential association between the six newly developed SSR markers and salinity tolerance in alfalfa.

References

Yu R, Wang G, Yu X, Li L, Li C, Song Y, Xu Z, Zhang J, Guan C. Assessing alfalfa (Medicago sativa L.) tolerance to salinity at seedling stage and screening of the salinity tolerance traits. Plant Biol. 2021; 23(4): 664–674.

Yu R, Wang X, Wang G, Xu Z, Gao Q, Du X, zhang Y. Analysis of salinity-tolerance and screening of salinity-tolerance evaluation indicators in Medicago sativa L. varieties at seedling stage. Acta Agrestia Sinica 2022; 30(7): 1781–1789.

Lin F, Li S, Wang K, Tian H, Gao J, Zhao Q, Du C. A leucine-rich repeat receptor-like kinase, OsSTLK, modulates salt tolerance in rice. Plant Sci. 2020; 296: 110465. 42.

Wang J, Liu S, Li C, Wang T, Zhang P, Chen K. PnLRR-RLK27, a novel leucine-rich repeats receptor-like protein kinase from the Antarctic moss Pohlia nutans, positively regulates salinity and oxidation-stress tolerance. Plos One. 2017; 12(2): e0172869.

Fan G, Gao Y, Wu X, Yu Y, Yao W, Jiang J, Liu H, Jiang T. Functional analysis of PagERF021 gene in salt stress tolerance in Populus alba × P. glandulosa. Plant Genome. 2024; 17(4): e20521.

Han X, Song C, Fang S, Wei Y, Tian J, Zheng X, Jiao J, Wang M, Zhang K, Hao P, et al. Systematic identification and analysis of the HSP70 genes reveals MdHSP70-38 enhanced salt tolerance in transgenic tobacco and apple. Int J Biol Macromol. 2025; 289: 138943.

Kang N, Cho C, Kim J. Inducible expression of Arabidopsis response regulator 22 (ARR22), a Type-C ARR, in transgenic Arabidopsis enhances drought and freezing tolerance. PLoS One. 2013; 8(11): e79248.

Shi Y, Tian S, Hou L, Huang X, Zhang X, Guo H, Yang S. Ethylene signaling negatively regulates freezing tolerance by repressing expression of CBF and type-A ARR genes in Arabidopsis. Plant Cell. 2012; 24: 2578-2595.

Krzywińska E, Kulik A, Bucholc M, Fernandez MA, Rodriguez PL, Dobrowolska GY. Protein phosphatase type 2C PP2CA together with ABI1 inhibits SnRK2.4 activity and regulates plant responses to salinity. Plant Signal Behav. 2016; 11(12): e1253647.

Xing B, Gu C, Zhang T, Zhang Q, Yu Q, Jiang J, Liu G. Functional study of BpPP2C1 revealed its role in salt stress in Betula platyphylla. Front Plant Sci. 2021; 11: 617635.

In order to make the description more accurate and logical, in lines 341-344, the“For instance, SSR231, SSR121 and SSR122, SSR121 and SSR125, SSR122 and SSR165, SSR122 and SSR191 can distinguish four different alfalfa varieties with 100% accuracy. SSR127 and SSR165 can distinguish LS from the other alfalfa varieties (Fig 7)”was moved to the Discussion section in the revised manuscript (Now in lines 456-459).

The “Transcriptome sequencing results showed that PP2C12, PP2C25, ERF026 in roots, and ARR5 in leaves were upregulated by NaCl tre

---

## [Editor Report · Decision Letter 1]

27 Oct 2025

Development of SSR markers related to salinity resistance based on transcriptomic sequences in Medicago sativa

PONE-D-25-38113R1

Dear Dr. Wang,

We’re pleased to inform you that your manuscript has been judged scientifically suitable for publication and will be formally accepted for publication once it meets all outstanding technical requirements.

Kind regards,

Mehdi Rahimi, Ph.D.

Academic Editor

PLOS ONE
---

## [Editor Report · Acceptance letter]

PONE-D-25-38113R1

PLOS ONE

Dear Dr. Wang,

I'm pleased to inform you that your manuscript has been deemed suitable for publication in PLOS ONE. Congratulations! Your manuscript is now being handed over to our production team.

Kind regards,

on behalf of

Associate Prof. Mehdi Rahimi

Academic Editor

PLOS ONE